# Potential Effects of Phenolic Compounds That Can Be Found in Olive Oil on Wound Healing

**DOI:** 10.3390/foods10071642

**Published:** 2021-07-15

**Authors:** Lucia Melguizo-Rodríguez, Elvira de Luna-Bertos, Javier Ramos-Torrecillas, Rebeca Illescas-Montesa, Victor Javier Costela-Ruiz, Olga García-Martínez

**Affiliations:** 1Biomedical Group (BIO277), Department of Nursing, Faculty of Health Sciences, University of Granada, Avda. Ilustración 60, 18016 Granada, Spain; luciamr@ugr.es (L.M.-R.); elviradlb@ugr.es (E.d.L.-B.); jrt@ugr.es (J.R.-T.); rebecaim@ugr.es (R.I.-M.); ogm@ugr.es (O.G.-M.); 2Institute of Biosanitary Research, ibs.Granada, C/Doctor Azpitarte 4, 4^a^ planta, 18012 Granada, Spain

**Keywords:** extra virgin olive oil, phenolic compounds, wound healing, tissue regeneration

## Abstract

The treatment of tissue damage produced by physical, chemical, or mechanical agents involves considerable direct and indirect costs to health care systems. Wound healing involves a series of molecular and cellular events aimed at repairing the defect in tissue integrity. These events can be favored by various natural agents, including the polyphenols in extra virgin olive oil (EVOO). The objective of this study was to review data on the potential effects of different phenolic compounds that can also be found in EVOO on wound healing and closure. Results of in vitro and animal studies demonstrate that polyphenols from different plant species, also present in EVOO, participate in different aspects of wound healing, accelerating this process through their anti-inflammatory, antioxidant, and antimicrobial properties and their stimulation of angiogenic activities required for granulation tissue formation and wound re-epithelialization. These results indicate the potential usefulness of EVOO phenolic compounds for wound treatment, either alone or in combination with other therapies. Human studies are warranted to verify this proposition.

## 1. Introduction

The treatment of wounds represents a considerable economic burden for health care systems worldwide. In the USA, 8.2 million individuals covered by “Medicare” were treated for wounds in 2014, at an estimated cost ranging between $28.1 billion and $96.8 billion [1]. In the UK, it was estimated that the treatment of wounds and associated complications cost the National Health Service a total of £5.3 billion in 2012/13, with £18.6 million spent on practice nurse visits, £10.9 million on community nurse visits, £7.7 million on physician visits, and £3.4 million on hospital outpatient visits [2].

A wound is defined as a loss of skin integrity and function produced by physical, chemical, or mechanical agents [3]. Wounds are classified as acute, i.e., healing within 7–10 days, or chronic, i.e., failing to close within this time period after a torpid and disordered healing process. Wound chronicity has been associated with various factors, including underlying disease, infection, prolonged inflammation, consumption of medications, and aging-related oxidative stress [4,5,6].

Wound healing is characterized by a series of molecular and cellular events aimed at repairing the defect in tissue integrity. This process can give rise to regeneration, in which specific lost tissue is replaced with parenchymatous cells of the same type, and/or repair, in which the tissue is replaced by non-differentiated elements of connective and support tissue, resulting in a scar [7]. Cellular and biochemical events in wound healing can be divided into inflammatory, proliferative, and remodeling phases. The inflammatory phase involves intense vascular activity characterized by: exudate production; blood coagulation; hemostasis; the release by platelets of growth factors and proinflammatory cytokines that attract leukocytes to the lesioned area; and the activation of immune cells (mastocytes, gamma-delta T cells, and Langerhans cells) that release cytokines and chemotactic factors that attract different cell populations [8,9,10,11]. This is followed by the proliferative phase, aimed at wound angiogenesis, fibroplasia, and re-epithelialization [12]. It is characterized by the migration and proliferation of: endothelial cells, responsible for the formation of new vessels; fibroblasts, responsible for the production of collagen and other extracellular matrix components needed to form granulation tissue; and keratinocytes, responsible for the formation of epithelium over the wound surface [12,13,14]. In the final remodeling phase, scar maturation is promoted through the substitution of type III collagen, widely present in granulation tissue, by type I collagen (COL-I), characteristic of the human dermis [15]. 

The wound healing process can be favored by the antioxidant, antimicrobial, and/or biostimulatory properties of certain natural agents. Among these, the positive effect of polyphenols on tissue regeneration in some vegetable species is well documented. For instance, it has been found that resveratrol, widely present in grapes and red wine, can control oxidative stress, stimulating wound healing and diminishing the inflammatory response [16]. Proanthocyanidins in berries and curcumin in turmeric have also been found to shorten the inflammatory phase and favor wound healing by reducing the production of tumor necrosis factor-α (TNFα), interleukuin-1 (IL-1), and/or reactive oxygen species (ROS) [17,18]. Quercetin, a flavonoid present in peppers, tomatoes, apples, and oranges, also possesses anti-inflammatory, antioxidant, and antimicrobial properties and has the capacity to stimulate the production of collagen and the proliferation of fibroblasts and keratinocytes, among other cell populations [19]. Polyphenols in extra virgin olive oil (EVOO) have also demonstrated beneficial effects in wound healing. However, only limited evidence is available on the mechanisms underlying the effects of EVOO polyphenols on tissue regeneration [20,21,22,23]. The objective of this study was to provide an updated review of published data on the impact of different phenolic compounds that can also be found in EVOO on wound healing. 

## 2. Results

EVOO is the only olive oil obtained by pressing and one of the few that undergo no additional refining process. It comprises triacylglycerols (~98%), fatty acids, and mono- and diacylglycerols. More than 230 minor compounds, notably phenolic compounds, constitute around 2% of the total weight of EVOO. It contains five main classes of phenolic compounds: flavonoids, lignans, phenolic alcohols, and secoiridoids [24].

Flavonoids, secoiridoids, phenolic acids, and phenolic alcohols are the main polyphenols found in olives. Flavonoids, phenolic acids, and phenolic alcohols are present in numerous fruits and vegetables from various botanical families, whereas secoiridoids are only found in plants of the Olearacea family, including the olive tree (*Olea europaea* L.) [25]. The main findings of this review are summarized in Table 1.

### 2.1. Flavonoids

In general, flavonoids are released in EVOO at the end of the maturation phase due to hydrolytic processes [92].

#### 2.1.1. Rutin

The flavonoid with the highest concentration in EVOO is rutin [92,93], which has been attributed with anti-inflammatory, antioxidant, and neuro-, nephro-, and hepato-protective properties [94,95,96]. In an in vivo model of tissue regeneration, the administration of rutin-supplemented hydrogels reduced the wound area in rats, attributed to a decrease in oxidative stress markers (e.g., lipid peroxidation); the production of carbonyl protein through the oxidation of certain amino-acids, including lysine, proline, or arginine; and/or a treatment-induced increase in protein levels [26]. In another rat study, the intraperitoneal administration of rutin proved effective in the treatment of diabetic ulcers, reducing the wound area and increasing the percentage closure by stimulating the production of antioxidant enzymes and inhibiting the expression of metalloproteinases and vascular endothelial growth factor (VEGF), thereby lowering oxidative stress and inflammatory responses [27]. In an in vitro study, Pivec et al. [28] demonstrated that the water-based enzymatic polymerization of rutin enhances its antioxidant activity, promoting the viability of fibroblasts and HaCat cells (non-tumoral human keratinocytes), evidencing its potential usefulness in tissue regeneration and wound healing. The regenerative effect of rutin has also been related to epidermal growth factor-controlled cytoplasmic and nuclear pathways and closely associated with the activity of keratinocytes with an important role in the response of epidermal cells to wounds [29]. The authors reported a dose-dependent reduction in monocyte chemoattractant protein-1 and inducible protein-10 and increase in IL-8 after treating human keratinocytes with rutin.

#### 2.1.2. Luteolin 

Luteolin has anti-inflammatory, antioxidant, and cell-protective properties and is present in various vegetable species, including EVOO [97,98,99]. Luteolin is known to affect multiple cell functions and to raise levels of E-cadherin, an important molecule for binding certain cells responsible for the integrity of animal tissue [30,34]. In an in vitro study, the treatment of cultured human platelets with luteolin influenced their coagulation and fibrin polymer formation, inhibiting procoagulant biomolecules such as fibrin and factor X [31]. Luteolin also reduces the initial inflammatory phase of wound healing and may prevent prolonged inflammatory processes, avoiding new local damage and a delay in wound closure [100]. This flavonoid also reduces the migration of leukocytes, the loss of plasma and reduction of edema, which may be of benefit after the initial immune response to a wound [101]. Luteolin and its derivatives are known to increase the proliferation of fibroblasts and/or keratinocytes, thereby accelerating wound closure [32,33]. In later phases of healing, luteolin can inhibit the activity of collagenase and hyaluronidase, enhancing the stability of the extracellular matrix [35].

#### 2.1.3. Apigenin

Like the other polyphenols belonging to the flavonoid group, apigenin (Api) has been extensively studied for its antioxidant properties [102]. However, its benefits are not only limited to its ability to neutralise ROS, but it has also been shown to modulate wound healing and healing processes. In vivo assays by Süntar et al. showed that Api induced an improvement in wound healing, with faster re-epithelialisation and higher collagen concentration compared to the control [36]. It appears that this effect could be related to the anti-inflammatory activity of Api, which was determined by Whittle’s method. Furthermore, in vitro assays developed by the same authors showed that Api is able to inhibit collagenase and hyaluronidase enzymes at a dose of 100 µg/mL, thus corroborating its regenerative potential in wound healing. In other in vitro studies previously developed by our research group, it was observed that Api at a dose of 10^−6^ M i stimulates the growth and migration of human fibroblasts in culture, showing an antimicrobial effect by inhibiting the growth of microorganisms of diverse nature, such as *Staphylococcus aureus, Staphylococcus epidermidis, Escherichia coli, Proteus* spp., and *Candida albicans* [37]. These antibacterial effects have also been documented by Cheng et al. according to which Api is able to inhibit the haemolysis of rabbit erythrocytes caused by *S. aureus*. Similarly, the application of a gel of Api and lysin LysGH15 showed bactericidal capacity against *S. aureus* and methicillin-resistant Staphylococcus aureus. These authors observed not only an antimicrobial effect, but also that this gel reduced the levels of proinflammatory cytokines accelerating wound healing in a mouse model [38]. Other authors have also studied the effectiveness of using Api-based gels for wound healing. This is the case of Shukla et al., who tested the benefits of this compound by making and applying a chitosan-based gel enriched with Api in an in vivo diabetic wound model and in a dead space wound model. These authors showed that catalase and superoxide dismutase levels were elevated in the group treated with the enriched gel, in addition to a greater antioxidant effect and increased healing compared to the group that was only treated with the chitosan gel [39]. Other studies carried out on recreated wound models in mice have shown that the topical application of an Api-based gel accelerates re-epithelialisation, reduces inflammatory processes, and promotes neovascularisation of the injured tissue [40].

### 2.2. Secoiridoids

Secoiridoids are characterized by the presence of an elenolic acid derivative in their molecular structure. Along with lignans, they are the most abundant phenolic compounds in EVOO.

#### Oleuropein

Oleuropein (OLE), a phenolic compound found in olive leaves, olives, and EVOO, has important antioxidant and anti-inflammatory properties [103]. OLE intervenes in dermal regenerative bioprocesses during wound healing, playing an anti-inflammatory role by reducing oxidative stress and lipopolysaccharide (LPS)-induced cell death [41,104,105,106] 

In vitro analysis of the biological effect of a phenolic compound concentrate from Olive Mill Wastewater, rich in OLE, showed an increase of more than 70% in the viability of HaCaT cells at 24 and 48 h post-treatment. Scratch assay results evidenced a significant increase in HaCaTcell migration versus controls, obtaining an 80% reduction in wound area at 18 h and complete closure at 24 h [42]. In addition, OLE treatment of human umbilical vein endothelial cell cultures (HUVEC-C) increased cell viability in comparison to non-treated controls [41]. In in vivo mouse studies of wound healing, intradermal OLE injections favored re-epithelialization by increasing the number of collagen fibers and upregulating VEGF protein expression [43,44].

Spanish researchers have patented various pharmaceutical formulae (gel, cream [o/w emulsion], or aqueous solution) that contain OLE as the sole active ingredient. Their application on diabetic foot, vascular, and pressure ulcers (PU) was found to be more effective than conventional treatments at all tested concentrations, with no significant differences as a function of treatment dose or pharmaceutical formulation. The best outcomes were obtained at the highest concentration used (10^−2^ M) [45].

### 2.3. Phenolic Acids

Phenolic acids are secondary metabolites of aromatic plants. They are present in small amounts in EVOO and are associated with its sensory and organoleptic qualities [107].

#### 2.3.1. Gallic Acid

Gallic acid (GA) can be obtained by the hydrolytic decomposition of tannic acid using a glycoprotein esterase [24,108]. It has been found to have antioxidant, antimicrobial, anti-inflammatory, and anticarcinogenic capacities as well as protective gastric, cardiac, neurological, and dermal effects [46,109].

With respect to the role of GA in wound regeneration, in vitro studies of HaCaT keratinocytes, MEF mouse embryonic fibroblasts, and HF21 human fibroblast cells reported that GA directly regulates the expression of antioxidant genes and accelerates the migration of the studied cell populations under both physiologic and hyperglycemic conditions. GA also activates factors that play a relevant role in wound healing, such as focal adhesion kinases, c-Jun N-terminal kinases, and kinases regulated by extracellular signals [47]. GA has been found to inhibit the proliferation, migration, invasion, and cell cycle progression of fibroblasts isolated from keloids, promoting their apoptosis, and these effects are likely mediated by inhibition of the AKT/ERK signaling pathway [48]. Differences between the aforementioned studies in the functions attributed to GA may be explained by the distinct origins and characteristics of the cell lines under study. 

With regard to its antimicrobial activity, GA has proven capable of inhibiting the motility, adhesion, and biofilm formation of multiple microorganisms involved in wound contamination, including *Pseudomona aeruginosa* and *Staphylococcus aureus* [110]. GA treatment has also been found to alter the cell membrane integrity of Gram-positive and Gram-negative bacteria, compromising the permeability of the bacterial membrane and increasing the accumulation of antibiotics within the microorganism [46,49,111].

Prompted by the above findings, researchers have recently focused on GA formulations based on chitosan to enhance its properties and bioavailability. In this way, Kaparekar et al. [50] developed a collagen-fibrin scaffold with GA-loaded chitosan nanoparticles (GA-CSNPs) to favor the sustained release of GA in the wound bed and expand its in vitro and in vivo bioavailability. Results in cell lines and experimental rats showed that the GA-CSNP scaffold increases the cell migration rate, accelerating wound contraction and shortening the epithelialization phase. Sun et al. [51] obtained encouraging wound healing outcomes in rats by applying a biocompatible antibacterial dressing based on chitosan-copper-GA.

#### 2.3.2. Vanillic Acid

Vanillic acid (VA) possesses antioxidant, antimicrobial, and antibacterial properties [52,112,113,114,115,116] and is widely used in the food industry as an aromatic, additive, and preservative [117,118,119,120,121]. With respect to its potential for wound healing, observations of the effect of VA (among other phenolic compounds) on human dermal fibroblast and epidermal keratinocyte cell cultures indicated that it can improve the healing of chronic wounds (e.g., venous ulcers or burns), exerting a protective effect against the free radicals derived from this type of lesion [53].

#### 2.3.3. Caffeic Acid

Caffeic acid (CA), which has antimicrobial, anti-inflammatory, antioxidant, anxiolytic, and antitumor properties, is present in fruit, vegetables, tea, and red wine as well as EVOO [105,122].

In 2008, Song et al. [54] observed that the treatment of skin incisions with CA progressively increased the levels of collagen-like polymers. In their in vitro study of NIH 3T3 fibroblasts, CA was found to exert antioxidant and anti-inflammatory effects by inhibiting ROS generation and releasing arachidonic acid and prostaglandin 2 (PGE-2). These results were verified by Romana-Souza et al. [57] in an in vivo pressure ulcer model.

Fibroblast proliferation is a key event in wound healing. It was significantly increased by treatment with 10-6M CA for 24 h, which upregulated the expression of COL-I, VEGF, platelet-derived growth factor (PDGF), fibroblast growth factor (FGF), and transforming growth factor-β (TGFβ), stimulating the migration of fibroblasts at 24 h [37]. In another study, treatment with 0.5 mg/mL CA for 1 min accelerated the fibroblast proliferative response in 3T3-L1 mice at 24 h, although this response was inhibited when the mice were treated for 32 min, likely due to a cytotoxic effect [123].

CA derivatives such as phenyl ester have also been reported to favor the healing of skin wounds, acute lesions, and severe burns in animal studies, reducing the anti-inflammatory response and oxidative damage and stimulating wound contraction and re-epithelialization [58,124]. Similar results have been obtained in in vitro models [55,56,58].

#### 2.3.4. Ferulic Acid

Ferulic acid (FA), which has demonstrated elevated antioxidant and anti-inflammatory capacities and antidiabetic, antitumor, and neuro- and cardio-protective properties [125,126], is found in vegetables and cereals in free or conjugated form [24,127,128,129].

FA has been proposed to have therapeutic potential in different phases of wound healing [130]. In vitro studies have shown that FA increases angiogenesis and neovascularization by stimulating VEGF- and PDGF-mediated signaling pathways, increasing endothelial cell and fibroblast proliferation and migration and potentiating the repair response [37,59,60,61]. In other studies, the synergic action of FA associated to other molecules or essential oils with biostimulatory activity increased the viability, migration, and proliferation of fibroblasts and keratinocytes, among other cell types [62,131,132]. 

In vivo studies in diabetic rats confirmed the promotion of wound healing by FA treatment, which reduced the time to closure [62,63,64]. The combined treatment of incisions in rats using FA extract and other phenolic compounds from the flowering plant Boerhavia diffusa for 14 days reduced the wound area by 14% [133]. In a mouse study, the application of ointments containing FA and other plant extracts on induced burns modulated the inflammatory response and enhanced wound regeneration [134]. FA-supplemented hydrogels were recently developed to stimulate wound healing and reduce closure time, obtaining favorable outcomes in animal studies [135,136].

#### 2.3.5. P-Coumaric Acid 

p-coumaric acid (pCA), also known as 4-hydroxycinnamic acid, has been reported to have antioxidant, anti-inflammatory, antitumor, and antimicrobial properties and to exert preventive effects against neurologic, nephrological, and hepatic toxicity and UVB radiation [137,138]. pCA is a phenolic acid from the family of hydroxycinnamic acids and is found in broccoli, tomatoes, carrots, spinach, beans, and olives, among others [24,127,129,139,140].

Various in vitro studies have demonstrated the therapeutic potential of pCA for skin wound healing. Melguizo-Rodríguez et al. [37] found that treatment with 50 µM pCA had regenerative effects, stimulating the growth, differentiation, and migration of cultured human fibroblasts. Likewise, treatment of the murine fibroblast line 3T3 with 3 or 30 µM pCA enhanced wound closure in vitro and had no cytotoxic effects [65]. pCA, among other phenolic acids from plants, has also shown antioxidant activity on fibroblasts and keratinocytes, enhancing its wound healing capacity [53,140,141]. However, the benefits of pCA for wound healing have not yet been investigated in vivo.

### 2.4. Phenolic Alcohols

Hydroxytyrosol (HTyr) and tyrosol (Tyr) are the main phenolic alcohols in EVOO [142,143]. Their concentrations are generally low in fresh oil but rise during its storage [144].

#### 2.4.1. Hydroxytyrosol

HTyr has been reported to have antiatherogenic, cardioprotective [145,146], anticancer [147,148], neuroprotective [149,150], antimicrobial [151,152], anti-inflammatory, and antiplatelet properties [66,153]. It is released by OLE hydrolysis, which also gives rise to OLE aglycone and elenolic acid [154].

HTyr can modulate cell signaling and therefore play an important role in the healing process [155]. In in vitro studies, HTyr was found to inhibit the production of nitric oxide and PGE-2 and the expression of cyclooxygenase-2 (COX-2) and metalloproteinase-9 and to increase the production of TNF-α in monocytes [66]. HTyr treatment has promoted the proliferation and migration of human keratinocytes, porcine vascular endothelial cells, and HUVEC-C cells, thereby favoring angiogenesis and the re-epithelialization of skin wounds [67,68,69]. The antioxidant activity of HTyr was demonstrated in human keratinocytes subjected to UVB rays, observing that it protects against DNA damage and reduces intracellular ROS [70]. HTyr treatment was also found to reduce β-galactosidase, responsible for senescence, in dermal fibroblasts subjected to UVA light-induced photoaging, suggesting a possible anti-aging effect on skin [84].

Healing can be delayed by the biofilm present in wounds, and HTyr has demonstrated antimicrobial properties against different bacterial strains as well as antiviral activity. Therefore, treatment with HTyr could contribute to wound healing by diminishing the microbial load in the wound bed [71,72,156].

#### 2.4.2. Tyrosol

Tyr occurs as such or in the form of elenolic acid esters [157] and is distinguished by its ability to maintain its antioxidant capacity even under critical conditions [73]. Besides EVOO, it can be found in numerous foods (meats, nuts, vegetables, and fruit, etc.). 

Various in vitro studies have demonstrated the antioxidant and anti-inflammatory capacities of Tyr in mononuclear cells of human peripheral blood (lymphocytes, monocytes, and macrophages). Culture of these cell populations in an inflammatory environment and their subsequent treatment with Tyr significantly reduced the secretion of proinflammatory cytokines (IL-1β and macrophage migration inhibitory factor) and inhibited ROS production and the phosphorylation of mitogen-activated protein kinase, which may regulate inflammatory processes in chronic wounds [74]. The intravenous injection of Tyr in rats previously inoculated with LPS reduced their TNF-α, PGE-2, and nitrous oxide levels, inhibited their expression of COX-2, and produced a dose-dependent translocation of nuclear factor kappa-light-chain-enhancer of activated B cells [76].

With respect to vascularization of the wound bed, treatment of rats with Tyr for six weeks improved oxygen transport, reduced plasmatic viscosity, and favored the brain capillary network. The resulting improvements in blood pressure and circulation would promote the formation of granulation tissue in wounds [77].

Tyr, like HTyr, exerts antimicrobial activity. In an in vitro study, the growth of different *Escherichia coli* strains was inhibited by treatment with Tyr alone or combined with HTyr and OLE through the inhibition of ATP synthase. It can therefore be useful against local wound infections or the biofilm itself [75].

### 2.5. Lignans

Lignans are phenolic polymers which, together with tannins, contribute to the flavour, aroma and colour of the oil. The amount of lignans present in virgin olive oil can be up to 100 mg/kg, but there are considerable variations between different varieties of oil. These include pinoresinol and its derivatives [158].

#### Pinoresinol

With regard to pinoresinol, the literature on its effects on wound healing is still very limited. A recent study showed that pinoresinol is able to stimulate the proliferation and migration of keratinocytes in culture, enhancing the re-epithelialisation of cell-free zones in a dose-dependent manner with better results at 10 uM concentration [78]. These results have also been observed in other cell lines, such as fibroblasts. Thus, Do et al. demonstrated that pinoresinol treatment significantly stimulates the migration of mouse embryo fibroblasts, apparently due to the activation of Gi-coupled receptors, which are closely related to the migratory processes of the aforementioned cells [79]. In addition, the antimicrobial effect of this compound has also been documented, which would favour healing processes and complete wound closure [80,81].

### 2.6. Other Compounds

#### Vanillin

Vanillin is an aromatic aldehyde with antioxidant, anti-inflammatory, and antiapoptotic properties [84,85,86,87,88,159,160,161]. It is widely used in food, cosmetic, and pharmaceutical industries and is also present in EVOO [162,163,164].

With regard to its role in tissue regeneration and wound healing, a vanillin-enriched hydrogel was used to treat mesenchymal stem cells derived from the adipose tissue of diabetic rats and upregulated the expression of markers of vascular regeneration, collagen deposition, and hair follicle reconstruction [83]. In another study, the application of vanillin-supplemented chitosan membranes on skin incisions in diabetic rats reduced the wound size and TNF-α and IL-1β levels, increasing re-epithelialization, angiogenic stimulus, and collagen deposition [89]. Different formulations of vanillin have also shown bactericidal and/or bacteriostatic activity against Gram-positive and Gram-negative bacteria [82,90]. All of the aforementioned effects of vanillin have also been documented in burns [91].

## 3. Conclusions

The results of in vitro and in vivo studies support the favorable effect of EVOO polyphenols on the healing of skin lesions, attributable to their anti-inflammatory, antioxidant, antimicrobial, and angiogenic properties. These compounds represent an interesting therapeutic option for wound healing when applied alone, in combination with other treatments, or in phenolic extracts, which are rich in multiple polyphenols that may exert synergic effects. However, clinical trials are required to verify these findings in humans and further elucidate the mechanisms underlying the action of these molecules.

## Figures and Tables

**Table 1 foods-10-01642-t001:** Summary of the role of phenolic compounds in wound healing.

EVOO Polyphenol Classification	Phenolic Compound	Clinical Relevance	Concentration Range	Reference
**FLAVONOIDS**	*Rutin*	In vitro Antioxidant and anti-inflammatory activity. Stimulation of cell viability of fibroblasts, HaCat cells and keratinocytes.	0.025 g (*w*/*w*)	[26]
			100 mg/kg body weight	[27]
		In vivo Reduction of wound area and increase in the rate of lesion closure through increased production of antioxidant enzymes lipid peroxidation or protein carbonyl peroxidation and decreased expression of oxidative stress markers and inflammatory processes.	0.5 g of rutin hydrate	[28]
			50 μM	[29]
	*Luteolin*	In vitro Elevates the expression of E-Cadherin which improve the generation of induced pluripotent stem cells. Inhibits procoagulant biomolecules Anti-oxidative and anti-inflammatory activities on keratinocytes and fibroblasts and immune cells Promotes fibroblast cells proliferation and migration.	7.5 μM	[30]
			25 μM	[31]
			Not specified	[32]
			1 μM	[33]
		In vivo Induces the expression of E-Cadherin and inhibits cancer metastases Faster wound healing.Reduction of edema and leukocyte migration in rats’ wounds. Inhibits hyaluronidase and collagenase activity in an in vivo wound healing assay.	20 μmol/L	[34]
			200 mg/kg	[35]
	*Apigenin*	In vitro Inhibition of collagenase and hyaluronidaseStimulates proliferation and migration of fibroblasts Antimicrobial properties	50 μg/mL	[36]
			10^−6^ M	[37]
			500 μg per 0.1 g ointment	[38]
		In vivo *Faster re-epithelialisation* *Increases levels of catalase and hyaluronidase* *Induces neovascularisation*	200 mg/kg	[36]
			500 μg per 0.1 g ointment	[38]
			30 mg	[39]
			0.2 g	[40]
**SECOIRIDOIDS**	*Oleuropein*	In vitro Increase of cell viability of HUVEC-C and HaCat cells and stimulation of migration of HaCat cells	50 μM	[41]
			1 and 0.1 mg/mL	[42]
		In vivo Anti-inflammatory effect. Promotes re-epithelialization by increasing collagen fibers and VEGF expression.	50 mg/kg	[43]
			50 mg/kg	[44]
		*Humans* More effective than conventional treatments in the management of diabetic foot, vascular and pressure ulcers	10^−1^ M–10^−11^ M	[45]
**PHENOLIC ACIDS**	*Gallic acid*	In vitro Antioxidant, anti-inflammatory and antimicrobial activity. Stimulation of cell proliferation and migration of fibroblasts (MET and HF21) and HaCat keratinocytes.	-	[46]
			10–200 μM	[47]
			1000 μM	[48]
			-	[49]
		In vivo Accelerates wound contraction and reduces epithelialization period combined with chitosan as a scaffold.	0.1–0.5 mg/mL	[50]
			40 mmol.L^−1^	[51]
	*Vanillic acid*	In vitro Antimicrobial effects Protection of cultured human skin cells against oxidative damage.	23–33 mM	[52]
			50, 100, 200, 500 μg/mL	[53]
	*Caffeic acid*	In vitro Antioxidant, anti-inflammatory and antimicrobial activity. Stimulation of cell proliferation, COL-1, VEGF, FGF, TGFβ and PDGF expression and migration of fibroblasts	10 mg/kg	[54]
			10^−6^ M	[37]
			10, 20, 30 µM	[55]
			10 μM	[56]
		In vivo Antioxidant, anti-inflammatory and antimicrobial activity. stimulation of wound contraction and re-epithelialization processes	10 mg/kg	[54]
			5 × 10^−6^ M	[57]
			10 μmol/kg	[58]
	*Ferulic acid*	In vitro Promotes the secretion of VEGF and PDGF and increases fibroblast proliferation and migration.	10^−6^ M	[37]
			10^−4^–10^−5^–10^−6^ M	[59]
			0.1–1–10 μg/mL	[60]
			10^−3^–10^−4^–10^−5^ M	[61]
		In vivo Reduces the healing time in diabetic rat wounds	1, 2, 5, 10 mg/mL	[62]
			10 mg/mL	[63]
			10 and 20 mg/kg	[64]
	*P-coumaric acid*	In vitro Stimulates growth, differentiation and migration of dermal fibroblasts and promotes wound closure of murine fibroblasts.	10^−6^ M	[37]
			30, 30 and 300 µM	[65]
**PHENOLIC** **ALCOHOLS**	*Hydroxytyrosol*	In vitro Antioxidant and anti-inflammatory activity. Promotes the proliferation of keratinocytes and endothelial cells, favoring the processes of angiogenesis and re-epithelialization. Antibacterial and antiviral activity	1–10 μmol/L	[66]
			0–100 µM	[67]
			1–5 µM	[68]
			10–100 µM	[69]
			25–100 µM	[70]
			1–10,000 µmol/L	[71]
			10–40 mg/Kg	[72]
	*Tyrosol*	In vitro Antioxidant and anti-inflammatory activity. Antibacterial activity	-	[73]
			0.5–1 µM	[74]
			0–35 mM	[75]
		In vivo Antioxidant and anti-inflammatory activity. Improves vascularization and circulation in rats	10–100 µM	[76]
			50 mg/kg daily i.g. for 6 weeks	[77]
**LIGNANS**	*Pinoresinol*	In vitro Stimulates proliferation and migration of keratinocytes and mouse embryo fibroblasts	10 µM	[78]
			10 µM	[79]
			15.2 and 10.9 µM	[80]
			100 µg and 2 mg/sensidisk	[81]
**OTHER** **COMPOUNDS**	*Vanillin*	In vitro Antimicrobial properties In vitro/in vivo Modulation of stem cells plasticity and therapeutic action in diabetic wound healing. Stimulates vascular regeneration, collagen deposition.	0.05 mol, 5.00 g	[82]
			25, 50, 70 µM	[83]
		In vivo Antimicrobial properties Protective effects against chronic mild stress induced in rats. Effective against bacteria in skin burns and promotion of epidermis regeneration and vascularization. Antimutagenic, antioxidant and anti-inflammatory properties. Antinociceptive properties in a visceral inflammatory model in mice. Reduce wound healing size, increase reepithelization, angiogenic stimulus, collagen deposition, reduction of levels of TNF-α, IL-1β, and increase of levels of IL-10, TNF-β and VEGF in diabetic rats.	40 mg/Kg	[84]
			10 and 40 mg/kg	[85]
			3.1–6.3–12.5–25 µM	[86]
			1–10 mg/Kg	[87]
			150 mg/kg	[88]
			192.2 mg, 126 mmol	[89]
			2.06 g, 10 mmol	[90]
			-	[91]

## Data Availability

Not applicable.

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
