# Peer review of "Potential Effects of Phenolic Compounds That Can Be Found in Olive Oil on Wound Healing"

_foods, 2021, doi:10.3390/foods10071642_

Round 1
Reviewer 1 Report
This review, is an interesting work and deals with a up-to-date and important topic with an impact on health. However, the authors should clarify that this is not a review on “the role of olive oil phenolic compounds on wound healing” but on “ the role of phenolic compounds that can be found in olive oil…”, as practically all referenced studies are not carried out with olive oil, but with other vegetable products or plants that contain these compounds, or with isolated compounds. There is an exception for the case of secoiridoids. As mentioned above, the review is interesting, but this aspect has to be clear in the title, abstract and introduction.
Lignans are one of the most abundant polyphenols in olive oil, which is even mentioned by the authors, namely pinoresinol. Therefore, it is not clear that there is no information or reference to these compounds, concerning their action on wound healing.
As the authors decided to divide the article into introduction, results and conclusions, even though it is not a systematic review, a chapter on methods would be interesting, explaining the keywords used, exclusion and inclusion criteria, etc.
Author Response
RESPONSE TO REVIEWER
REVIEWER 1
Thank you very much for your comments which will contribute to complement the content and improve the quality of our work.
- This review, is an interesting work and deals with a up-to-date and important topic with an impact on health. However, the authors should clarify that this is not a review on “the role of olive oil phenolic compounds on wound healing” but on “ the role of phenolic compounds that can be found in olive oil…”, as practically all referenced studies are not carried out with olive oil, but with other vegetable products or plants that contain these compounds, or with isolated compounds. There is an exception for the case of secoiridoids. As mentioned above, the review is interesting, but this aspect has to be clear in the title, abstract and introduction.
Response: Thank you very much for your suggestion. Indeed, as the reviewer points out, most of the studies used in this review come from other plants. This is because the scientific evidence on the use of phenolic compounds extracted from olive oil in wound healing is limited. This is why, through this review, we aimed to highlight the potential of olive oil polyphenols for the treatment of wounds, based on the results obtained in other plant species that contain the same compounds as olive oil.
Thus, as suggested by the reviewer, we have modified the title of the manuscript to avoid misunderstandings.
“Potential effects of phenolic compounds that can be found in olive oil on wound healing”
We have also clarified the issue raised by the reviewer in both the abstract and the introduction.
Line 15-18: “The objective of this study was to review data on the main potential effects of different phenolic compounds that can also be found in EVOO on wound healing and closure. Results of in vitro and animal studies demonstrate that EVOO polyphenols from different plant species, also present in EVOO…”
Line 72-75: “The objective of this study was to provide an updated review of published data on the impact of EVOO phenolic compounds different phenolic compounds that can also be found in EVOO on wound healing.”
- Lignans are one of the most abundant polyphenols in olive oil, which is even mentioned by the authors, namely pinoresinol. Therefore, it is not clear that there is no information or reference to these compounds, concerning their action on wound healing.
Response: As suggested by the reviewer, we have completed the section "lignans" with the effects of pinoresinol on wound healing.
“2.5. Lignans
Lignans are phenolic polymers which, together with tannins, contribute to the flavour, aroma and colour of the oil. The amount of lignans present in virgin olive oil can be up to 100 mg/kg, but there are considerable variations between different varieties of oil. These include pinoresinol and its derivatives [140].
2.5.1. Pinoresinol
With regard to pinoresinol, the literature on its effects on wound healing is still very limited. A recent study showed that pinoresinol is able to stimulate the proliferation and migration of keratinocytes in culture, enhancing the re-epithelialisation of cell-free zones in a dose-dependent manner with better results at 10uM concentration [141]. These results have also been observed in other cell lines, such as fibroblasts. Thus, Do et al. demon-strated that pinoresinol treatment significantly stimulates the migration of mouse embryo fibroblasts, apparently due to the activation of Gi-coupled receptors, which are closely related to the migratory processes of the aforementioned cells [142]. In addition, the an-timicrobial effect of this compound has also been documented, which would favour healing processes and complete wound closure [143,144].”
- As the authors decided to divide the article into introduction, results and conclusions, even though it is not a systematic review, a chapter on methods would be interesting, explaining the keywords used, exclusion and inclusion criteria, etc.
Response: Thank you for your insights. The existing literature on wound healing includes studies with a great diversity of methodologies, including assays on different cell populations and species, as well as the use of compounds extracted from plant species of different nature. These reasons make it difficult to develop a systematic review. We have therefore carried out a narrative review to provide a more comprehensive overview of the results published to date on the use of phenolic compounds in wound healing. Please note that we have followed as a model the distribution of other narrative reviews recently published in Foods:
- What Is the Relationship between the Presence of Volatile Organic Compounds in Food and Drink Products and Multisensory Flavour Perception?. Foods 2021, 10(7), 1570; https://doi.org/10.3390/foods10071570
- Curcumin: a review of its effects on human health. Foods 2017, 6 (10):92
- Encapsulation and Protection of Omega-3-Rich Fish Oils Using Food-Grade Delivery Systems. Foods 2021, 10(7), 1566; https://doi.org/10.3390/foods10071566
In the same way, this distribution model is followed in the narrative reviews published by other journals:
- Application of Salivary Biomarkers in the Diagnosis of Fibromyalgia. Diagnostics 2021, 11(1), 63; https://doi.org/10.3390/diagnostics11010063
- Stimulation of brown adipose tissue by polyphenols in extra virgin olive oil. Critical Reviews in Food Science and Nutrition. 2020 July. Doi: 1080/10408398.2020.1799930
- Vitamin D and autoimmune diseases. Life Sci. 2019 Sep 15;233:116744.doi: 10.1016/j.lfs.2019.116744. Epub 2019 Aug 8.
- Salivary Biomarkers and Their Application in the Diagnosis and Monitoring of the Most Common Oral Pathologies. Int J Mol Sci. 2020 Jul 21;21(14):5173.doi: 10.3390/ijms21145173.
- SARS-CoV-2 infection: The role of cytokines in COVID-19 disease. Cytokine Growth Factor Rev. 2020 Aug;54:62-75. doi: 10.

Reviewer 2 Report
This review describes the effects of the polyphenols found in extra virgin olive oil (EVOO) on wound healing. This process is characterized by a series of molecular and cellular events aimed at repairing the tissue integrity defect.
The authors then underlined the importance and clinical relevance of these studies.
However, some criticisms persist. Olive oil, which is characterized by a high content of oleic acid, has been shown to exert a synergistic effect with the components present in different extracts.
Based on the quantitative and qualitative differences of the EVOO composition, which affect its biological activities, the olive oil study should specify the oils used.
Alternatively, in Table 1, a range could be indicated for each component responsible for clinically relevant effects.
Author Response
RESPONSE TO REVIEWER
REVIEWER 2
We would like to thank reviewer 2 for his input on our work. We have made the suggested modifications that will undoubtedly contribute to the improvement of this manuscript.
- This review describes the effects of the polyphenols found in extra virgin olive oil (EVOO) on wound healing. This process is characterized by a series of molecular and cellular events aimed at repairing the tissue integrity defect.
The authors then underlined the importance and clinical relevance of these studies.
However, some criticisms persist. Olive oil, which is characterized by a high content of oleic acid, has been shown to exert a synergistic effect with the components present in different extracts. Based on the quantitative and qualitative differences of the EVOO composition, which affect its biological activities, the olive oil study should specify the oils used.
Response: Thank you very much for your comments. Indeed, as you point out, the phenolic compound content of the oil varies depending, among other factors, on the variety of olive. This was observed in previous studies by our research group on human osteoblast cultures together with the synergistic effect that may occur between the compounds present in the oil as you point out*. Our interest in exploring the clinical applicability of this type of compounds in other tissues has led us to identify the potential effects of polyphenols in wound healing as an objective of this work. Due to the complexity of this process, the published scientific literature is heterogeneous in terms of methodology and nature of the compounds used. In this sense, the evidence on the effect of phenolic compounds extracted from oil and their application in wounds is still very limited. Therefore, in this study we have incorporated publications that use compounds present in various plant species that can also be found in olive oil. This work thus raises the need to develop tests to corroborate the findings of this review with compounds extracted directly from olive oil.
* Phenolic Compounds in Extra Virgin Olive Oil Stimulate Human Osteoblastic Cell Proliferation. PLoS One. 2016 Mar 1;11(3):e0150045. doi: 10.1371/journal.pone.0150045. eCollection 2016
- Alternatively, in Table 1, a range could be indicated for each component responsible for clinically relevant effects.
Response: We have included a new column in the table specifying the doses used in each job.
